# “Pandemic Fatigue! It’s Been Going On since March 2020”: A Photovoice Study of the Experiences of BIPOC Older Adults and Frontline Healthcare Workers during the Pandemic

**DOI:** 10.3390/healthcare10101967

**Published:** 2022-10-08

**Authors:** Angela U. Ekwonye, Abigail Malek, Tenzin Chonyi, Stephanie Nguyen, Valerie Ponce-Diaz, Lisa Lau Haller, Iqra Farah, Mary Hearst

**Affiliations:** 1Department of Public Health, St. Catherine University, 2004 Randolph Ave, St. Paul, MN 55105, USA; 2School of Nursing, University of Minnesota, Minneapolis, MN 55454, USA

**Keywords:** pandemic fatigue, BIPOC older adults, BIPOC frontline healthcare workers, challenges, self-care practices

## Abstract

The coronavirus disease 2019 (COVID-19) pandemic intensified the stressful and already difficult circumstances of communities of color. Yet, there is no current photovoice research highlighting the lived experiences of these communities from two perspectives—the older adults (OAs) and the frontline healthcare workers (FLHWs). This qualitative study used photovoice to visually portray the struggles of Black, Indigenous, and persons of color (BIPOC) OAs (*n* = 7) and younger FLHWs (*n* = 5) who worked with older adults during the pandemic and how they coped and recovered from the challenges of the pandemic. The investigators conducted a three-day training of ten research assistants (RAs) who were paired with either an OA or an FLHW for the photovoice sessions conducted in four stages. Upon examination of the narratives, focus group transcriptions, and photo stories, it became clear that participants faced different challenges during the pandemic, such as the fear of COVID-19 exposure, struggles to adopt COVID-19 mitigation strategies, workplace challenges, and social isolation. Amid this crisis of suffering, isolation, and sadness, participants employed two major strategies to deal with the challenges of the pandemic: positive reappraisal and self-care practices. The findings have implications for clinical social workers, mental health counselors, faith communities, nurse managers and administrators, and policymakers.

## 1. Introduction

The coronavirus disease 2019 (COVID-19) pandemic illuminated and intensified the long-standing and glaring inequities in healthcare utilization [1,2], education [3], mental health [4], socioeconomic indicators [5], and community resources [6,7,8] for older Black, Indigenous, and persons of color (BIPOC) and inadequate protections for BIPOC frontline healthcare workers who serve older adults in the United States [9,10]. It is untenable to separate the COVID-19 pandemic, subsequent isolation [11,12,13], and the lived experience of BIPOC. There is strong evidence that racial/ethnic minority and underserved groups, especially BIPOC communities, are disproportionately affected by COVID-19 [14]. Cases of COVID-19, hospitalizations due to COVID-19, and COVID-19 mortality rates are inordinately elevated in BIPOC communities [15,16]. Frontline healthcare workers are also at increased risk of contracting COVID-19 compared to the general community, and that risk doubles among healthcare workers of color [10]. Moreover, BIPOC healthcare workers face not only the risk of infection but also low wages and potential biases of patients within their care [17,18,19,20]. This is concerning because of the substantial increase in women of color in the direct care workforce [21,22].

Given these glaring disparities, it is imperative to understand the lived experience of BIPOC older adults and frontline workers grappling with COVID-19, isolation, racial bias, and mental health issues in order to develop policies during and after the pandemic that strengthen—not devitalize—these populations. This qualitative study used photovoice [23] to visually portray the struggles of BIPOC adults over age 65 and BIPOC frontline healthcare workers who worked with older adults during the COVID-19 pandemic and how they coped and recovered from the challenges of the pandemic.

### 1.1. Impact of the Ongoing Pandemic on BIPOC Older Adults and Frontline Healthcare Workers

The ongoing COVID-19 pandemic has led to substantial community-wide disruption, especially among communities of color already experiencing disproportionate poverty, crime, unemployment, racism, and discrimination rates [24]. Analyses of federal, state, and local data throughout the COVID-19 pandemic show that people of color have experienced a disproportionate burden of cases and deaths [25]. Data from the Centers for Disease Control and Prevention [CDC] showed there are large disparities in COVID-19 hospitalizations for American Indian and Alaska Native (AIAN), Black, and Hispanic people, although the overall disparities in age-adjusted risk for infection, hospitalization, and death have narrowed over time [26].

The rapid spread of COVID-19 among communities of color has been attributed to systemic racism, home living conditions, lack of access to health care, pre-existing mistrust of the public health and medical system, lower wage-earning jobs deemed ‘essential workers’, and more [1,6,8,24,27]. The lower-wage-earning individuals in these communities often cannot work from home during the COVID-19 crisis and may include frontline healthcare workers employed in hospitals and long-term care facilities [9,28]. For some, travel to work involves crowded public transportation, putting them at greater risk of contracting the virus [27]. During this pandemic, older persons of color have reported higher rates of stressors and emotional distress than their White counterparts [29]. The pandemic intensified the stressful and already difficult circumstances of communities of color who reported high stress, anxiety and/or depression, and social isolation [30]. People are at risk of psychological harm when kept in isolation, but the most vulnerable in these situations include older adults, minority groups, and those from lower socioeconomic groups [11,12,13,31,32,33]. There is concern that mental health symptoms from the COVID-19 pandemic may persist for many years, especially among communities of color who already receive less mental health treatment [34,35]. Communities of color will likely experience disproportionate mental health impacts of COVID-19 from exposure, cumulative burden, and social isolation [36]. This research aimed to capture the struggles of BIPOC older adults and frontline healthcare workers during the COVID-19 pandemic and how they constructed meaning using photovoice methodology.

### 1.2. Why Use Photovoice?

Photovoice is an innovative method when compared to the epidemiological approaches that guide most current discourse around COVID-19. Photovoice allows participants to document their experiences using photography [23]. The photovoice methodology used in this study was based on Paulo Freire’s theory of critical consciousness, which assumes that oppression is a reality worldwide, and it is in people’s best interest to fight to remove it from their midst [37]. When individuals exercise critical consciousness, they explore questions related to human dignity, freedom, authority, social responsibility, and personal purpose. Such awareness can enhance a sense of personal and social responsibility and empowerment. In this study, participants portrayed their experience of racism, isolation, and meaning during the COVID-19 pandemic using visual images as an educational tool for critical consciousness and subsequent reflection on how they coped with the challenges. Current data highlight inequities for BIPOC older adults and healthcare workers but provide no opportunity for critical discourse to change future policies. As an interpretive method, photovoice has an epistemological advantage over positivist research, because it allows researchers to ask different questions and understand lived experiences through a socially and culturally constructed lens. By putting the tool in the hands of the participants to bring visual images of their struggle during the ongoing COVID-19 pandemic, this method allowed participants to voice their interpretation of the images through the interviews and focus group discussions. This study aimed to understand and elevate the socially and culturally constructed experiences of the COVID-19 pandemic among BIPOC older adults and frontline healthcare workers.

### 1.3. The Current Study

To date, no current photovoice research highlights the struggles of BIPOC older adults and frontline healthcare workers during the COVID-19 pandemic and how they constructed meaning in a negative event. While photovoice research has been conducted with older adults [38,39,40,41,42], no published research has explored the struggles of communities of color from two perspectives—older adults and younger frontline healthcare workers and how they found meaning during the pandemic. Challenges of the pandemic (isolation, quarantine) can threaten personal meaning, a significant source of life satisfaction, personal growth, and healing [4]. There is evidence that meaning has significance in how people make sense of their lives, adjust to certain stressors, and integrate their life stories in ways that help them lead fulfilled and meaningful lives [43]. Yet, no study has examined how BIPOC older adults and healthcare workers construct meanings in the face of the challenges of COVID-19. By exploring how this minority group experiences and constructs meaning using visual images, we could learn more about how well they respond to opportunities and manage problems in their lives.

## 2. Materials and Methods

### 2.1. Design

The study was a descriptive qualitative design using photovoice conducted between January and May 2022. The photovoice method allowed for the active involvement of the participants. It enabled researchers to perceive the world from the perspective of participants. The approach allows participants to reflect on the issue, engage in critical dialogue through small group discussion of photographs, provide meaningful interpretations of their photos, and share their viewpoints with others [23].

### 2.2. Participants

Participants for this research were BIPOC older adults (OAs) and frontline healthcare workers (FLHWs) who worked/work with older adults. Purposive sampling was used to identify BIPOC older adults. Inclusion criteria include older adults aged 65 years or older living independently and alone in a house or apartment. The adult must be able to consent for themselves, identify as BIPOC, be physically able to take photographs, and attend (virtually or in-person) the focus group discussion. Since we wanted to capture their experience of isolation during the pandemic, we excluded older adults living with family members in nursing homes and assisted-living facilities. Older adults who could not consent for themselves, physically take photographs, or participate in the focus group sessions were also excluded. Recruitment materials were available in English, Spanish, Somali, and Vietnamese. Fliers were posted in local independent living facilities, high-rise public housing, local non-profit organizations, community centers, and religious institutions in the Minneapolis–St. Paul Area. Frontline healthcare workers were recruited through the University’s School of Health, long-term care facilities, and non-profit organizations serving older adults. Participants were included if they were current students at any degree level and working with or caring for older adults within the past year. Fliers for the FLHWs were also posted at different places on campus. Interested participants were asked to contact the investigators if they were interested in participating in the study. In the end, seven OAs and five FLHWs were enrolled in the study. Our final sample size of twelve was based on the information power of the sample participants. Research indicates that the larger the sample’s information power, the lower the N required [44]. The authors [44] suggested that a study will need the least number of participants when the study aim is narrow, if the combination of participants is highly specific to the study aim, if the study is supported by established theory, if the interview dialogue is strong, and if the analysis includes an in-depth exploration of narratives. We posit that our sample size of twelve adequately captured participants’ experiences of racism, isolation, and meaning during the COVID-19 pandemic and secured representation for the study population for several reasons: (1) Our study aim was narrowly focused on participants’ experiences of racism, isolation, and meaning. (2) Participants were of a specific target group (BIPOC). (3) Our study was based on Paulo Freire’s theory of critical consciousness. (4) There was strong, clear, and frequent communication and support between research assistants (RAs) and participants. (5) We conducted an in-depth analysis of the data. The average age for OAs was 71 ± 7.35 years and 21 ± 4.03 years for FLHWs. The demographic characteristics of the participants are displayed in Table 1.

### 2.3. Training of Research Assistants (RAs)

The investigators conducted a three-day training of ten RAs during which the investigators explained the purpose of the research, the photovoice methodology, the use of the camera, and ethical photography. Details of the training are displayed in Table 2. After the training, the RAs were paired with either an OA or a FLHW for the photovoice sessions, which were conducted in four stages.

### 2.4. Procedure

#### 2.4.1. Stage 1: Introduction of Photovoice Methodology

Each RA contacted their assigned participant via phone call and set up a meeting date. During the first meeting, the RA explained the research purpose, questions, and photovoice methodology and obtained consent from each participant. In the second meeting, the RAs demonstrated the camera use, explained ethical photography, and had the participants practice using the camera, including troubleshooting. The RAs reminded each participant to obtain a waiver from any individual in their photographs.

#### 2.4.2. Stage 2: Taking the Photos

The RAs provided digital cameras to participants. Some participants were allowed to use their phone cameras to accommodate their preferences. The RAs gave their participants one week to take photos of (1) people, situations, or objects that reflect any challenges they faced during the COVID-19 pandemic, and (2) photos of people, situations, or objects that reflect the ways they sought meaning during those challenges. Should the participants decide to take photos of themselves, the researchers instructed the RAs to inform them to ask any of their relatives or friends to take the pictures. During the week of photography, the RAs remained in contact with their matched participant(s) via phone call. They identified and met the level of support the participants needed and provided verbal and written reminders of the intended focus of the photographs.

#### 2.4.3. Stage 3: Photo Selection Process

Once participants finished taking photos, each RA conducted and recorded a semi-structured interview. The participant was shown their photographs on a laptop or tablet and asked to choose two photos of their challenges during the COVID-19 pandemic and two photos of how they found meaning during those challenges. To help narrow down the most relevant photos, the RAs asked the participants questions such as “why did they take the photo?”, “how did they feel during the process?”, “what does the photo represent?”, and “how does this answer the research question?”. Once two photos that represented challenges and two photos that represented meaning were selected, the RAs supported participants during the writing of the narratives for each photo. To document the participant’s narrative of the images, the RAs used the SHOWED technique asking each participant: (1) What do you SEE in the picture? (2) What HAPPENED, or is HAPPENING, in the picture? (3) How does this relate to OUR lives? (4) WHY does the problem, concern, or strength exist? (5) What can we DO about it [23]? Each RA created a file with the selected photographs and narrative.

#### 2.4.4. Stage 4: Focus Group Discussion

The research team conducted two virtual focus group discussion sessions with older adults and one with frontline healthcare workers. One older adult focus group session was conducted in Spanish. During the focus group sessions, each participant was allowed to share and discuss their selected images of the challenges they faced and the images of how they found meaning. After each participant had finished discussing what the images meant to them, other focus group members were asked to reflect on the images and narratives and share their views. This guided discussion was used to deepen individual narratives of photographs, elevate the participant’s perspectives to researchers and policymakers, give community voice, and bring critical consciousness to future research questions, policies, and interventions [23]. The photo selection and narratives were recorded with the RA’s phone, while the focus group discussion sessions were recorded in Zoom. Each participant was awarded a $150 gift card for their time and participation. The study was approved by the investigators’ University Institutional Review Board (IRB) Protocol number 1624, and all subjects gave informed consent before participating.

### 2.5. Data Analysis

The interpretive, phenomenological approach was used to understand the lived experience of BIPOC older adults and frontline healthcare workers during the pandemic [45]. Recordings were transcribed using the software otter.ai. The study team reviewed transcripts for clarity. Interviews conducted in Spanish were transcribed and translated into English by a Spanish-speaking graduate student. The participants’ photographs interpreted by the participants were paired with the narrative. Data analysis was conducted in five phases by the interpretive team consisting of two lead investigators, who are experts in the content area of inquiry, and five RAs. In phase 1, the team reviewed, discussed, and interpreted the transcripts of participants’ initial narratives. Missing or unclear pieces were explored and discussed. Lines of inquiry resulting from initial interpretations guided subsequent interviews. In phase 2, the team identified central concerns and meanings that unfolded for specific participants. The team discussed life episodes that affected participants, events shaping their experiences of COVID-19 struggles, and how they found meaning. We explored the relationships between each participant’s experiences and meaning. The team wrote, reviewed, and revised summaries of central concerns with salient excerpts for each participant’s story. The team identified parts of stories or instances with similar meanings within participants’ stories, particularly vibrant stories (paradigm cases) that were compelling. In phase 3, the team identified and compared the shared meaning and central concerns of participants. In phase 4, the team developed in-depth interpretations of excerpts, central concern and meaning summaries, and interpretive summaries. In phase 5, the research team reported on the interpretations of central concerns and meaning, found in the results section, including some examples of participants’ photos and quotes. Throughout this process, the interpretation continued to be iterative to avoid losing sight of each participant’s particular story and context.

## 3. Results

### 3.1. Challenges of the COVID-19 Pandemic

Upon examination of the narratives, focus group transcriptions, and photo stories, it became clear that participants faced different challenges during the pandemic, such as (1) fear of COVID-19 exposure, (2) struggles adopting COVID-19 mitigation strategies, (3) workplace challenges, and (4) social isolation.

#### 3.1.1. Fear of COVID-19 Exposure

Both older adults and frontline healthcare workers were afraid of contracting COVID-19. Three of the five FLHWs (60%) perceived personal vulnerability to COVID-19 exposure at the workplace and the possibility of dying from it despite taking all precautions. One FLHW narrated how the emergency department (ED) was a potential site for COVID-19 exposure.

“The picture (Figure 1) shows you how the room you are going into is a yellow zone which indicates a potential to contract COVID-19. Whoever goes into the room has a higher chance of getting COVID-19; whenever we go in, we don’t know the potential risks, and we don’t know who we will come across. Many loved ones have entered the ED for something we did not see as COVID-19 and could not get back to their loved ones.”[FLWH 03]

**Figure 1 healthcare-10-01967-f001:**
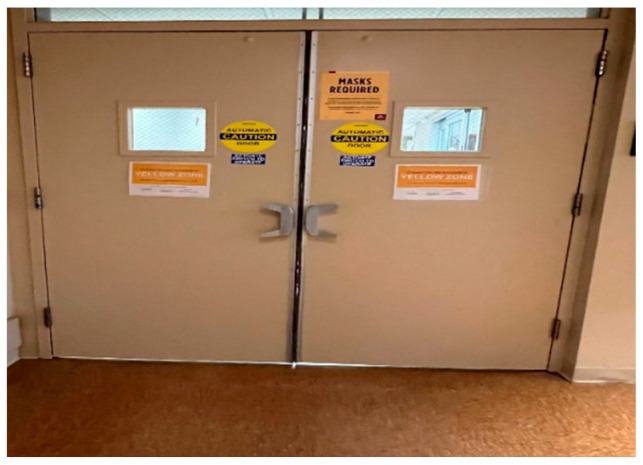
The entrance to the emergency department (ED).

While FLHWs’ fear stemmed from perceived susceptibility to contracting COVID-19 at the workplace, the OAs were more worried about the severity of the infection due to their age. Some emphasized the need for older people to take extra precautions. “As an elderly, you got to be extra careful because with the COVID-19 infection… severity differs… but as an older person, you are more at risk for serious consequences. So you have to be extra careful.” [OA 17].

#### 3.1.2. Struggles Adopting COVID-19 Mitigation Measures

Both OAs and FLHWs described the challenges in following the COVID-19 protocols. There were mixed attitudes toward wearing a mask, physical distancing, and hand sanitizer use. Although FLHWs and OAs recognized the protective role of wearing masks and practicing hand hygiene, both groups shared their frustration about the constant use of personal protective equipment (PPEs). One FLHW shared how healthcare workers had to mask up during the eight-hour work shift and, in some cases, for twelve hours. She, however, shared her frustration at the unwillingness of the general public to wear masks even for a few hours to help reduce the spread of the virus.

“Frontline healthcare workers use it for eight hours, eight straight hours. We only can take it off when we go for a break or when we’re alone, but we almost have to wear it the whole time. People complain about using masks for 30 min to go to the grocery store. We have to use it all the time when interacting with patients, walking in the halls, walking into the building. We must wear it because it’s now part of our uniform. People should just wear masks to help protect others; if we can do it for eight hours or even more for those that work for twelve hours, people should not be reluctant to use them.”[FLHW 03]

It seemed that FLHWs were more accepting of the constant use of masks because of the protection they believed it conferred. In contrast, FLWHs noted resentment from OAs when it came to the use of PPEs. They reported that OAs needed constant reminders to wear masks and use hand sanitizer.

“Not everyone likes wearing masks, especially working with elderly people at home care; they don’t always agree with the COVID-19 protections we are taking. About half of the elderly don’t like the mask rules, but they just have to be reminded that it is for their protection. We let them know that these precautions are being taken in different settings around the world and in casual settings like work. Just keep reminding those that we are working with in a home why we are taking these precautions and keep practicing the precautions.”[FLHW 01]

“Because of the pandemic, like with the masks, hand sanitizers are also being used a lot. Hand sanitizers can be annoying to have to use constantly. That could also be a problem in-home care, especially if we go out with our older adult clients. Say, like they have a baseball event or something, you have to constantly remind them to “oh, make sure you are using your hand sanitizer”. Sometimes that can be a bit difficult and frustrating for them, but it’s not too much difficult, but it’s just the precautions that have to be taken as much as we can.”[FLHW 06]

OAs and FLHWs echoed the experience of emotional and physical fatigue associated with following the ongoing COVID-19 mitigation measures.

“Pandemic fatigue! It’s been going on since March 2020. A lot of us are sick. It’s been literally years of social distancing, and isolation, quarantining, testing, vaccinations, and discussions about that, and wearing masks.”[FLHW 11]

An OA who quarantined several times during the pandemic narrated the difficult experience with these words:

“I see the staircase down to the basement (Figure 2) where I quarantined a couple many times during this COVID-19 of more than two years and a half, you know, more than two years. As a nurse, you know, I’ve been working with COVID-19 patients, on and off, though not very serious patients, you know, despite all of the PPE you wear, there’s every chance of exposure, so I always go down to quarantine for 4–5 days. That just reminds one of the difficult periods of this pandemic. It was mentally difficult.”[OA 17]

**Figure 2 healthcare-10-01967-f002:**
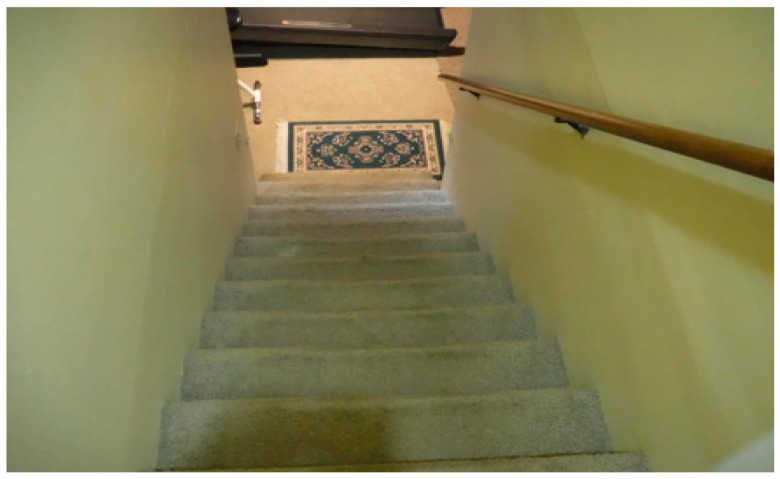
The staircase to the basement where one older adult quarantined a couple of times during the pandemic.

#### 3.1.3. Workplaces Challenges

The COVID-19 pandemic exacerbated existing workplace challenges and discrimination, with an exceptionally emotional and physical toll on BIPOC health care workers. Staff shortages were one of the workplace challenges mentioned by the FLHWs, which they noted contributed to the feeling of being burned out from work situations that were emotionally demanding.

“There have been mornings where I’m by myself for the first few hours and have to take care of about 40 residents. Although it’s technically doable, as a residency assistant, I normally get about 20 residents to care for. So it’d be about 40 residents by myself, which can be stressful.”[FLHW 03]

“You know, we, as healthcare workers, need breaks as well. There’s like such a high level of burnout and turnover in the medical field; people are just straight up quitting and retiring early, leaving the profession completely. So, like, we’re already stressed. And then we’re short-staffed. And so, it’s like, having to manage our own sanity and health amidst this crazy thing (COVID-19 pandemic), you know, is really difficult.”[FLHW 11]

The FLHWs who constantly wore masks at the workplace expressed the difficulties they experienced in getting masks at the beginning, which eased out as the pandemic progressed.

“When the pandemic started, we didn’t have as many supplies. But now that we have built up the supplies, we are using a lot of single-use masks. We are trying our best with our PPE, which is one of the best ways to prevent it from patient to provider, providers to other providers.”[FLHW 11]

The FLHWs also complained about the inadequate compensation and bonuses despite the heavy workload and health risks from working with COVID-19 patients. “I work at an assisted living place. Additionally, we work with COVID-19 positive residents and don’t get additional pay. When we go into those rooms, it is like exposing ourselves willingly.” Another FLHW described how the lack of flexibility in workplace rules (rigid structures, Figure 3) limited her from interacting and bonding with residents, which is one of the things she loves to do at work.

“We have that set of rules, those set of things that we have to do, which makes it hard to do what I really love about my job, which is the talking and communicating with others. So the majority of my time, I’m doing med passes, and I’m giving my residents their medication. Then I’m just simply moving on to the next place.”[FLHW 01]

The FLHWs alluded to the lack of empathy, understanding, and respect from the leadership, particularly when they fell short of completing their assigned work despite the heavy work overload.

“I have a nurse who is very organized and structured but has never been a Certified Nursing Assistant before. The nurse will call and yell at us over the phone for not being at the right place at the right time. I distinctly remember being with a resident and getting a call asking where I was. Then, the nurse was upset with me that I was behind on my schedule, despite being short-staffed. The first thing the nurse thought to do in that situation was to be argumentative instead of understanding the situation. The lack of understanding or the feeling of not being respected, especially during that stressful time, was painful.”[FLHW 01]

#### 3.1.4. Social Isolation

Participants described social isolation as being away from their loved ones. The isolation was particularly hard for older adults, who described it using words and phrases such as suffering, loss of friends, stuck inside, etc. One older adult shared: “The pandemic was very, very hard on us, because we had to isolate ourselves. A person like us who are elderly, it is very hard to be isolated because we need to have people around us.” [OA02]. The feeling of isolation triggered sadness in some older adults. One older adult sometimes cried in response to the deep sadness of being isolated from her loved ones.

“This is the window (Figure 4) that when my children came, they looked there. They left some little things there, and they left. It was very sad in here. They only entered the porch. I only looked out the window looking at them, it took a long time, it took me a long time. I was very sad because when my children came before the pandemic, they came in, and we ate here at the table. And that was no longer there. So, I suffered because they couldn’t get in. They said because they could infect me. What can I say? I felt bad. I felt bad because they couldn’t come in as usual and talk. They just came from a distance and left. Sometimes I even cried because my children could not come in. That suffering was very hard for me, if not for everyone; we distanced ourselves for a long time.”[OA 01]

**Figure 4 healthcare-10-01967-f004:**
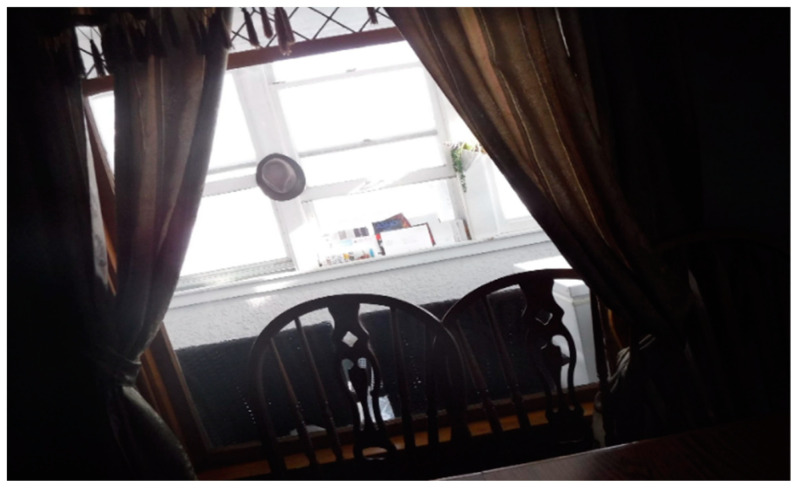
A photo of a window that an older adult looked through to see her loved ones when they came to visit.

### 3.2. Overcoming the Challenges of the COVID-19 Pandemic

The COVID-19 pandemic created many challenges for BIPOC OAs and FLHWs who were afraid of contracting the virus and dying. In the midst of this crisis of suffering, isolation, and sadness, participants found ways to cope and recover from the challenging experiences of the pandemic. Participants used two major strategies: positive reappraisal and self-care practices, to deal with the challenges of the pandemic.

### 3.3. Positive Reappraisal

Positive appraisal refers to cognitive strategies used to assess an event in a favorable light leading to the perceived benefit from the stressful event [46]. The FLHWs particularly viewed the pandemic positively to help lessen the burden of compliance with COVID-19 mitigation measures and workplace challenges. They were able to reduce the associated negative emotions of the experiences by finding humor in a COVID-19 poster, motivational quotes, and rhymes. One FLHW shared a funny picture that was hung up in their work area to get everyone’s attention about wearing masks and, at the same time, lighten the challenges of the pandemic.

“It’s hung up in our work area (Figure 5). The photo shows a fox along with some rhymes promoting the use of masks. With the pandemic, there are mask mandates everywhere so masks are a big part of our lives. The fact that there is a doodle along with the rhymes just makes it more lighthearted. It reminds me a little of Dr. Seuss, with all of the rhymes that just automatically lighten the darkness the pandemic has brought. It encourages people to wear masks in a different way that isn’t so demanding. I guess continuing to lighten the mask situation for others can really help others who are struggling with wearing masks.”[FLHW 06]

**Figure 5 healthcare-10-01967-f005:**
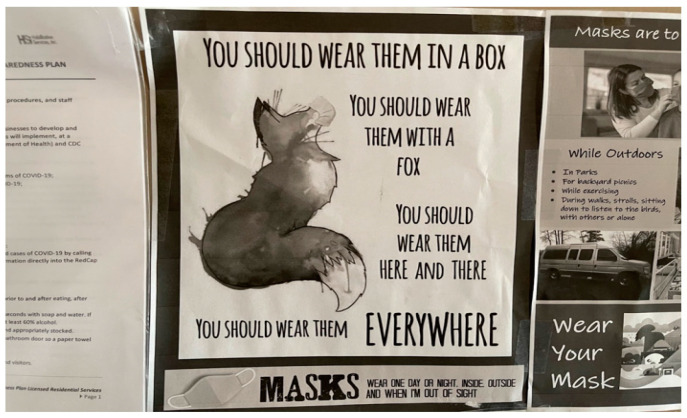
This funny poster was used to remind others to wear their mask.

Another FLHW who shared a photo of the first COVID-19 test in France in 1553 described how she distracted the patients with her jokes to help lighten the mood and potentially relieve the COVID-19 swab’s discomfort. Picture not shown.

“Here is a funny picture that my co-worker posted to help bring laughter to patients getting the COVID-19 test. It sparks conversation and serves as a bit of distraction and a little joke, saying it could be worse. Back in the day, my co-worker used it to say, ‘this is how the COVID-19 test was done in the 1550s’, as a joke. The public must get COVID-19 tests regularly, like a vaccine, to keep us all safe. Although the process is painful, still, we have to endure it because this could have been worse. We can always try to find the light in the dark tunnel. We can find a burst of laughter out of it.”[FLHW 03]

### 3.4. Self-Care Practices

Self-care generally refers to the process of purposeful engagement in practices and activities that promote holistic health and well-being of the self or provide stress relief [47]. Participants engaged in emotional, spiritual, social, and physical self-care practices to cope or recover from the challenges of the pandemic.

#### 3.4.1. Emotional Self-Care

Emotional self-care includes different practices to cope with stress and reduce emotional damage from negative experiences. Emotional self-care activities can take the form of creative arts expression. Participants found ways to support their emotional health through making music, crafts, baking, coloring, and reading. One older adult found purpose, meaning, and solace throughout the pandemic by connecting with nature, artistically expressing herself through drumming, and focusing on things that she enjoyed and found meaningful.

“I am standing in front of a tree (Figure 6), out in nature, completely immersed in my energy work (more specifically, the energy ball) and holding my drum in the image to the right. I have found purpose, meaning, and solace throughout the pandemic by centering myself in good energy, nature, practices that ground myself and others, artistic expression through staying active in my community, and through my musical expression of drumming. Despite the chaos of the world around me due to the pandemic, I am able to ground myself in my passions. Life is often centered around meaning. My aim is to provide hope through these images, that despite such a difficult and catastrophic event such as this pandemic, people can still find meaning and joy through things and hobbies that they love. Suppose more people begin to focus on the things that they enjoy. In that case, it may help alleviate some of the stress and anxiety that has clearly resulted from the public crisis at hand.”[OA 03]

**Figure 6 healthcare-10-01967-f006:**
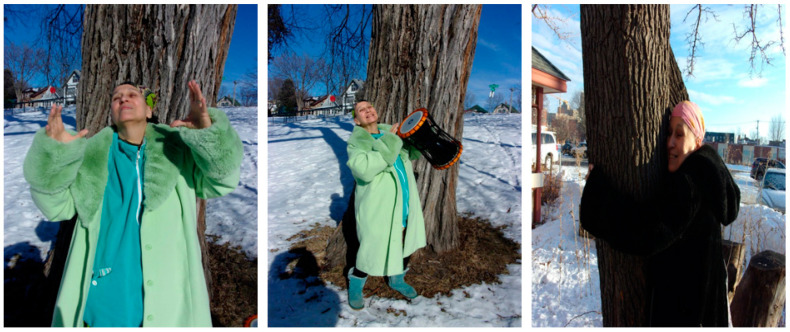
This trio of images show an older adult not only full of joy, but finding rejuvenation.

Another older adult engaged in creative artistic expression such as coloring Easter eggs and decorating cookies (Figure 7) to cope with the boredom of the COVID-19 pandemic.

“I participate in the elder lodge family on Easter by coloring eggs and decorating cookies. It brings us together as a family and keeping us active with what we have been doing in our lives with our children. It helps us be involved with the outside celebrations. It keeps our elder family strong by participating together. It brings the elder lodge family together. We enjoy group activities, and sometimes we do crafts. Just make sure that we keep these activities alive and it helps strengthen our minds. We have to concentrate on crafts.” [OA 10]

One participant found reading particularly helpful in combatting the pain of isolation during the pandemic.

“I see myself sitting, you know, there behind the table… I am just starting to read. This was especially helpful during the pandemic and especially in the winter times when you cannot go out and do any gardening. So reading takes you to a different world even if you’re in quarantine and sitting alone, yet you are not alone.”[OA 17]

#### 3.4.2. Spiritual Self-Care

Spirituality is the aspect of humanity that refers to how individuals seek and express meaning and purpose and how they experience their connectedness to the moment, to self, to others, to nature, and to the significant or sacred [48]. Spiritual self-care involves nurturing connections and finding meaning in life [49]. Participants engaged in spiritual self-care practices such as quiet reflection, prayer and meditation, service to others, and practicing gratitude. One participant described how her connection to the moment through quiet reflection led to a greater understanding of self and acceptance of her limitations.

“I do a lot of reflection about my time on the job because I absolutely love my residents and like you’re at this heightened sense of awareness. It’s a lot of understanding, knowing my own limitations, and my limitations in relation to things that I cannot change.”[FLHW 01]

A different participant discussed the usefulness of taking a break to rest and reflect.

“This is Minnehaha Falls in Minneapolis (Figure 8), and the waterfall is frozen, creating a cool icicle effect. You can see the branches and the water freezing over and see that it is winter. I, my daughter, and my wife were enjoying the outside, bundled up, and my daughter was on a little sled. Taking walks like this is a respite from pandemic fatigue and the day-to-day existence of living inside a global pandemic. So, it’s a nice little break from that. In terms of pandemic fatigue, it’s been going on since March of 2020… it’s been literally years of social distancing, isolation, quarantine, testing, vaccinations, wearing masks, and all of the discussions that revolve around it. Especially working in the healthcare field, it’s been taking its toll of having to work and seeing people affected by that. Getting back into enjoying nature and understanding that nature still exists outside of this very human problem is important, and it’s helpful to take a break, rest, and reflect to get to one’s self.”[FLHW 11]

**Figure 8 healthcare-10-01967-f008:**
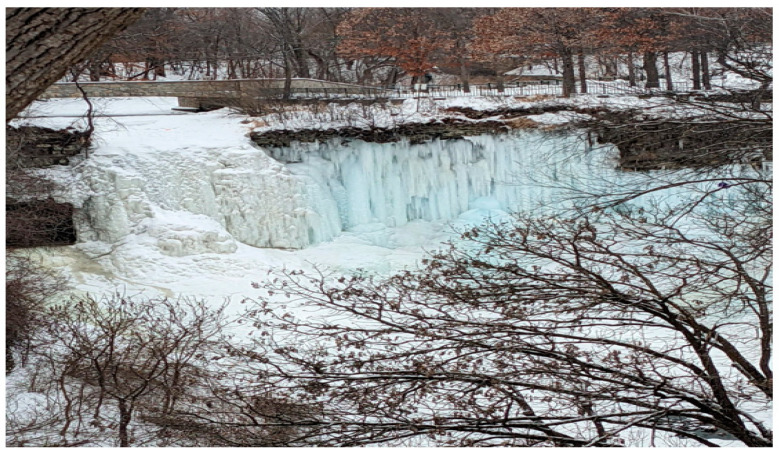
Minnehaha Falls in Minneapolis, Minnesota represented a place for quiet reflection.

“Gratitude flows from a heightened sense of awareness, increasing as one takes the time to go within and be quiet. Expression of gratitude was another important source of strength for participants, who described it as the feeling of thankfulness for God’s protection and appreciation of others’ kindness and generosity.

My son goes out to the street for his first job working with children. In his second job, he works with clients, and thank God we have not had any type of infection because among the three of us who live in this house we take great care of ourselves. So, we are fine and happily healthy, which is the most important.”[OA 02]

“In the medical facilities, we’re wearing cloth masks. It was really inspiring to see so many community members donating masks and stuff.”[FLHW 11]

In the healing process, people can become entirely absorbed in changing the effects of negative situations in their life. If individuals remain focused on themselves, they will not be fully connected to the world. Participants appeared to have realized that a critical part of healing is being of service to others. The BIPOC OAs expressed their deep inner drive to do something that engenders hope in others.

“I am a member of a group that provides aid for those outside their community, such as food, clothes, and furniture. I want to present some of the people behind the work of my church (picture not shown) who provides food, clothing, household appliances, and furniture. I love being a part of their church. It is part of what I have been doing all my life to ensure everything gets delivered—especially the food and clothing. Besides a spiritual need, there is a need for food and clothing. Still, now there is a line for these food shelves. It can give them hope that there are people out there that have the means to distribute food and clothing to other reservations. We can pray for them to keep their work growing and expanding.”[OA 10]

Despite how depressing the pandemic seemed, participants were optimistic that things would improve in the future.

“The staff in the hospital took the time to write this quote (Figure 9), and when other staff come in through a specific location, they see this quote and bring it with them throughout their day. Yes, this pandemic sucks, but it will get better. Sometimes, we go through so many hardships and want to give up, and this quote motivates us to keep going. The quote is a strength to help encourage people, and it gives a few words of encouragement. Most of us are overworked, and we don’t get recognition from our managers, and just saying this always inspires and motivates us. And it’s hard to keep your head up, even though it has been only two years, but it could be three or four. It will get better.”[FLHW 03]

#### 3.4.3. Social Self-Care

Social support is an essential coping mechanism during times of crisis. Social self-care involves participation in activities or spending time with peers, family, and patients and the meaning found in those interactions. BIPOC older adults appeared to have relieved their feelings of social isolation by participating in book clubs, Easter activities, and connecting with loved ones. One participant shared how her connections with her granddaughter brought her meaning and joy despite feeling isolated from many of her friends (Figure 10).

“We are both smiling deeply and very happy to be with one another. The relationship between us emulates unconditional love and that we care about each other dearly. My granddaughter provides help to me by serving as my Personal Care Assistant. Not only this, but she also both gives and receives that familial love, aiding me as my constant companion and close friend. I have been able to see my granddaughter in person throughout the entirety of the pandemic. Despite being isolated from so many people, being able to share love and time with her has brought me so much meaning and lots of joy despite such a fearful time. Since the onset of the COVID-19 pandemic, many of my in-person interactions have ceased to exist due to safety precautions. However, this image shows my ability to stay close in connection with a few individuals despite feeling isolated from a great number. Until the safety of the virus is clear, finding solace in time with my close loved ones and those in my circle is key.”[OA 03]

While OAs mostly found meaning through connecting with loved ones and engaging in group activities, the FLHWs found meaning through interactions with their patients.

“I prefer doing more care over medicine, and some people prefer the opposite. I like the care because I can interact, communicate, talk to my clients, and actually get to know my residents and better bond with them.”[FLHW 01]

#### 3.4.4. Physical Self-Care

Emotional and mental well-being, especially during times of crisis, can depend on finding ways to dissipate the negative effects of those stressors physically. Participants described how they engaged in bicycling, walking, hiking, snowshoeing, sledding, tai chi, crossword puzzles, and gardening to promote relaxation and reduce stress levels. Older adult participants described how they turned to gardening to promote their emotional well-being during the pandemic. One said: “I was gardening, you know, during the COVID-19. I was stuck in the house. I needed something to do”. [OA 14].

Another shared: “Since during the COVID-19 times, you cannot go out and socialize much, you cannot go out in the community gathering. It was difficult to go to meet your friends and relatives here in the United States, or you know, elsewhere; gardening was one the best and most rewarding experiences, you know, especially during this pandemic. This one beautiful white tulip (Figure 11) is a sign of beauty, a sign of strength, and a sign of one’s ability to face all the difficult challenges. So that gives me a lot of courage and also happiness”. [OA 17].

An older male participant shared how he turned to crossword puzzles to help relieve stress and improve cognitive functioning.

“I found a way how to work my brain through crossword problems. I can do four crossword problems in two hours. This keeps my brain awake and the development of my personality. I concentrated so much on my crossword puzzles to help my brain develop and focus. Doing this type of mental exercise and other physical activities helped me have a better form of life and quality. It made my mind more agile and awake and helped me a lot.”

## 4. Discussion

The COVID-19 pandemic has been and continues to disproportionately affect BIPOC older adults and frontline healthcare workers who serve older adults in the United States [9,10]. The purpose of this photovoice study was to provide an opportunity for BIPOC older adults (OAs) and the frontline healthcare workers (FLHWs) to portray their struggles and how they coped and recovered from the challenging experience of the pandemic, in order to develop policies during and after the pandemic that will strengthen these populations.

The findings from our study indicate that participants feared the potential risk of COVID-19 exposure and death. Their fear of COVID-19 exposure and death seemed valid, considering that rates of COVID-19 infection and mortality nationwide were disproportionately higher for BIPOC healthcare workers [50,51]. The FLHWs, in particular, expressed the difficulties they encountered obtaining PPE at the beginning of the pandemic. However, this became easier as time went on. Both OAs and FLHWs felt frustrated with the prolonged use of PPEs, although the FLHWs were more accepting of COVID-19 policies. Most FLHWs experienced burnout from the excessive workload, emotional exhaustion, the feeling of unappreciation, inadequate compensation, and the lack of empathy and respect from their bosses. Participants’ challenges were consistent with the findings of other studies which demonstrated that FLHWs faced PPE shortages [51], extended work hours due to chronic understaffing [52,53], inadequate wages [54], and dwindling emotional reserves to support patients [55,56].

The COVID-19 policies of social distancing and quarantine, designed to stop the rapid spread of infections, exacerbated participants’ feelings of isolation and emotional distress. The sense of isolation was more challenging for OAs, who had to distance themselves from their loved ones because of their increased vulnerability to COVID-19 infection. Our older adult participants lived independently and alone in a house or apartment, increasing the risk of social isolation. This finding corroborated the results of a previous study of older adults in the Minneapolis metropolitan area where the study participants were recruited [57]. Quarantine increased isolation among the older participants, consistent with previous research [58]. It is well known that social disconnectedness can exacerbate perceived isolation and lead to an increased risk of depression and anxiety among older adults, while social participation is a key component for health and quality of life in aging adults [59,60].

To combat the challenges of the pandemic, participants engaged in two primary coping mechanisms: positive reappraisal and self-care practices. A positive appraisal is a form of meaning-based coping that enables individuals to adapt successfully to stressful life events [61]. The need to find meaning in adverse life events plays a crucial role in coping and resilience building during significant negative events [43]. The FLHWs mainly saw the pandemic positively by finding some amusements in a COVID-19 poster, deriving emotional relief from motivational quotes, and cracking COVID-19 jokes to make light of the situation. Positive reappraisal of negative life events has also been shown to enhance the perception of a positive outlook on life [46]. They also reported the experience of discrimination, such as being underpaid, ignored, disrespected, etc. It appeared that these positive reappraisal strategies helped to relieve the negative effects of the workplace’s discriminatory experiences. The usefulness of positive reappraisal in managing emotions in the face of discrimination events was demonstrated in a previous study [62]. Although BIPOC older adults in this study did not mention the use of positive reappraisal as a strategy in dealing with the challenges of COVID-19, previous research indicates that positive reappraisal was associated with better life satisfaction, lower perceived stress, and improved mental and physical health for older adults [63,64].

Participants generally supported their overall health and well-being through engaging in emotional, spiritual, social, and physical self-care practices. The role of self-care practices in improving people’s well-being during the COVID-19 lockdown was evident in a previous study [65]. The older adult participants combatted the pain of isolation caused by COVID-19 policies of social distancing, isolation, and quarantine by engaging in different creative arts such as making music and crafts, baking, coloring, and reading. Creative arts expression has been shown to increase self-awareness, reduce anxiety levels, improve mood, and reduce feelings of isolation and alienation for individuals in times of high stress [66,67], which the COVID-19 pandemic exemplified. Having outlets to express oneself allows one to look through the layers of his or her own creative expression to find healing through art and creativity. While emotional self-care practices were evident among the older adult participants, none of the frontline healthcare workers mentioned using creative arts activities to cope with the COVID-19 pandemic challenges. Balancing the many responsibilities of their roles may have resulted in little time for the caregivers to engage in any creative arts activities of their own.

OAs and FLHWs overwhelmingly engaged in spiritual self-care practices such as self-reflection, prayer and meditation, service to others, and practicing gratitude to connect to the moment, self, others, nature, and the significant or sacred. Research shows that finding and basking in the present moment whenever one is feeling stressed or worried or experiencing negative emotions can help one return to a calm, peaceful state of mind [49]. The majority of the FLHWs reported spending quiet time in self-reflection which allowed them to become more aware of their inner selves. Such conscious connection and stillness can bring people into a place of expanding peace, love, and healing [49]. Some participants engaged in prayer and meditation to mitigate the challenges of the pandemic. Research has demonstrated the usefulness of prayer and meditation in alleviating job stress and burnout and enhancing overall well-being [68,69]. Focusing on oneself can limit the full potential to heal. It seems the BIPOC older adults in the present study realized that in order to heal, they must be of service to others, exhibiting that pure act of love and connectedness and giving others hope, consistent with the literature [4].

While older adult participants derived some emotional relief participating in group activities and spending time with loved ones, FLHWs found meaning spending time with their older adult patients. Research shows that interacting with a social network provides individuals with opportunities for enjoyment, support, and encouragement [4]. Other studies have demonstrated the role of meaningful social interactions in mitigating social isolation and negative mental health outcomes and bolstering psychological resilience for frontline healthcare workers [70] and older adults [71] during the COVID-19 pandemic. Participants engaged in different forms of physical activities such as bicycling, walking, hiking, snowshoeing, sledding, tai chi, and gardening to promote relaxation and reduce the stress associated with the pandemic. Previous research has demonstrated the usefulness of physical activities in fighting against the mental, emotional, and physical consequences of the COVID-19 quarantine [72]. The practice of gardening, which was popular among the older adult participants, seemed to have improved their endurance and strength, reduced stress levels, and promoted relaxation. The physical and mental health benefits of physical activities for older adults and frontline healthcare workers during the pandemic were consistent with other studies [73,74].

### 4.1. Strengths, Limitations, and Opportunities for Future Research

Given the substantial community-wide disruption caused by the COVID-19 pandemic, especially among communities of color already experiencing disproportionate rates of poverty, crime, unemployment, racism, and discrimination, the research findings highlighted the lived experiences of BIPOC older adults and the frontline healthcare workers who work/worked with older adults during this COVID-19 pandemic. The study allowed us to put a camera in the hands of the participants to bring visual images that emphasize the unique challenges they faced during the pandemic. The study also enabled us to examine how BIPOC older adults and frontline healthcare workers construct meaning in the face of adversity. Despite the strengths of our contributions, our study is not without limitations. First, the sample size was small (*n* = 12), so the findings cannot be generalized to the broader population. We suggest that future research consider increasing the sample size for greater external validity. Second, participants were paired with different RAs, so we cannot be sure of the uniformity of the protocol. However, to reduce this problem, all RAs were trained on the research protocol, and the investigators had a weekly meeting to ensure protocol adherence. Third, some interviews were conducted in person and others virtually, which might have limited the quality of interaction between the RAs and the participants. Future studies should ensure uniformity in the study protocol for increased internal validity.

### 4.2. Implications for Practice

This study revealed BIPOC older adults’ and frontline healthcare workers’ challenges during the pandemic and how they coped and recovered from the challenging experience. The findings will be relevant to clinical social workers, mental health counselors, faith communities, and local non-governmental organizations that can learn more about the capacity of BIPOC OAs and FLHWs to respond to adversity and manage problems in their lives. This knowledge can give them distinctive insights into the processes of positive adaptation for BIPOC OAs and FLHWs so they can better support them in finding meaning during negative life events and in any future pandemic. The FLHWs voiced their dissatisfaction and lack of flexibility and support from their administrators, emphasizing the need for human resources policies that allow for a flexible and person-centered approach in the workplace. Furthermore, nurse managers and administrators should promote a supportive workplace environment by providing psychosocial support and integrating resilience education/training, which may help buffer workplace stress to improve the job satisfaction of BIPOC frontline healthcare workers. Finally, both OAs and FLHWs engaged in different cognitive, behavioral, and cultural practices to build resilience, hence suggesting the need for policymakers to pay attention to these factors that shape the experiences of communities of color. Such understanding will enable them to enact policies that will increase access to resilience resources for this population.

## 5. Conclusions

To the best of our knowledge, no current photovoice research has highlighted the experience of the challenges of the COVID-19 pandemic from two perspectives—the older adults and the frontline healthcare workers and how they constructed meaning during the pandemic. This study has provided insights into the lived experience of the participants in dealing with the challenges they encountered during the pandemic. It is clear that to reduce inequities in health, greater attention needs to be paid to understanding the world views of the BIPOC communities and how they make sense of adverse events in their lives, which is critical to their psychological functioning and overall well-being.

## Figures and Tables

**Figure 3 healthcare-10-01967-f003:**
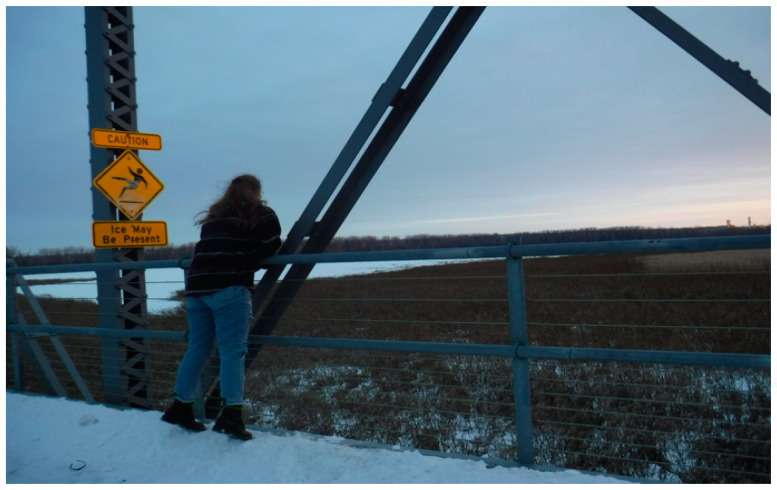
The structure with the beams in the photo was used to illustrate the rigidity of workplace rules.

**Figure 7 healthcare-10-01967-f007:**
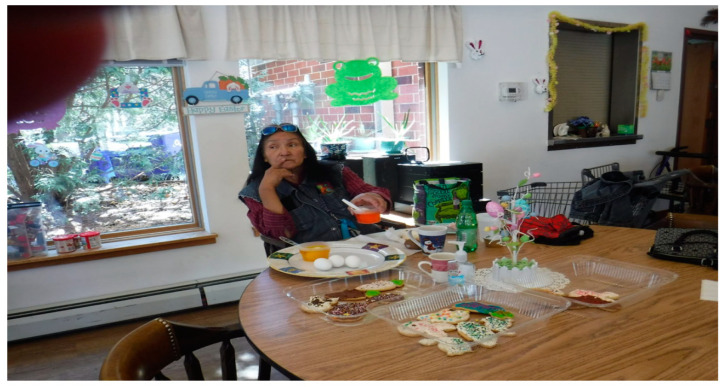
A photo of an older adult coloring Easter eggs.

**Figure 9 healthcare-10-01967-f009:**
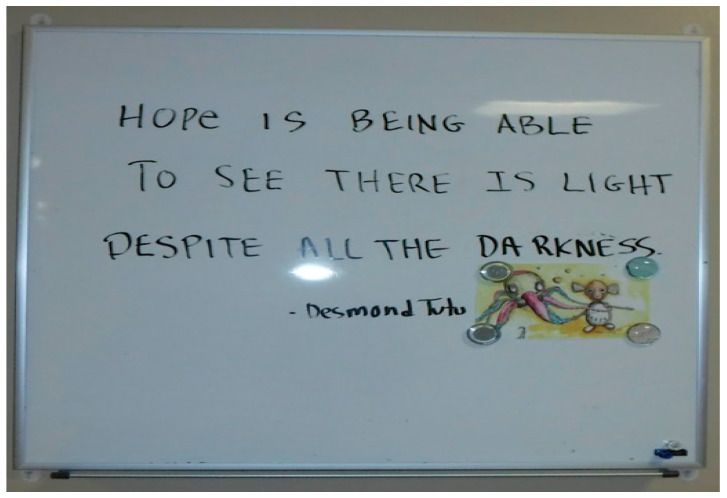
This was a quote that someone wrote that brought inspiration to possibly everyone who passed through that area.

**Figure 10 healthcare-10-01967-f010:**
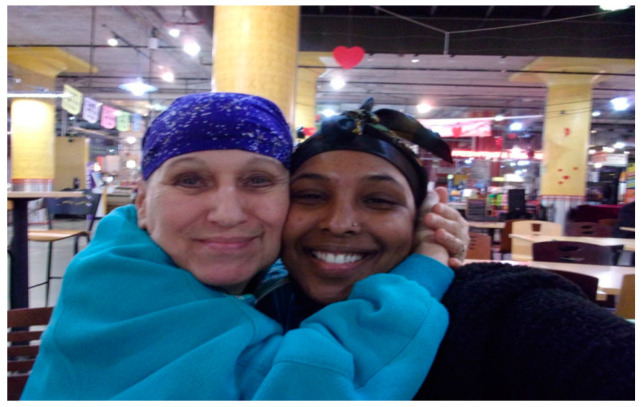
“In this image, my granddaughter and I are sharing an embrace.”

**Figure 11 healthcare-10-01967-f011:**
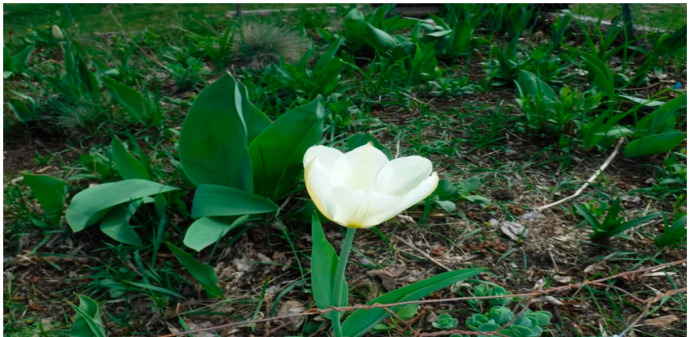
A photo of a garden.

**Table 1 healthcare-10-01967-t001:** Demographic description of Black, Indigenous, and persons of color (BIPOC) older adults (OAs) (*n* = 7) and frontline healthcare workers (FLHWs) (*n* = 5).

Demographics	OAs *n* (%)	FLHWs *n* (%)
Gender		
Female	6 (86%)	4 (80%)
Male	1 (14%)	1 (20%)
Race/Ethnicity		
Hispanic or Latino	5 (71%)	2 (40%)
Asian	1 (14%)	2 (40%)
Native America	1 (14%)	
Black/African American		1 (20%)
Education		
Some Secondary school	1 (14%	
Some College	4 (57%)	
Undergraduate	1 (14%)	
Post-graduate	1 (14%)	
Freshman		1 (20%)
Sophomore		1 (20%)
Junior		2 (40%)
Graduate		1 (20%)
Major		
Public Health		2 (40%)
Pre-nursing		1 (20%)
Public Policy		1 (20%)
Masters of Public Health		1 (20%)
First generation college student		
Yes		2 (40%)
No		3 (60%)
Relationship status		
Single (Never married/divorced)	5 (71%)	3 (60%)
Married	2 (29%)	1 (20%)
Did not answer		1 (20%)
Occupation		
Retired	3 (43%)	
Nanny	1 (14%)	
Nurse	1 (14%)	
Clergy	1 (14%)	
Unemployed	1 (14%)	
Religion		
Catholic	2 (29%)	1 (20%)
Evangelist	1 (14%)	
Christian	2 (29%)	
Buddhism	1 (14%)	
Agnostic	1 (14%)	2 (40%)
Islam		1 (20%)
None		1 (20%)
Born in USA		
Yes	3 (43%)	3 (60%)
No	4 (57%)	2 (40%)
Primary language		
Spanish	4 (57%)	1 (20%)
English	2 (29%)	4 (80%)
Tibetan	1 (14%)	

OAs = Older adults. FLHWs = Frontline healthcare workers.

**Table 2 healthcare-10-01967-t002:** Details of the training sessions with research assistants.

Session	Content	Objectives	Assessment
Day 1: 3 h	Study overview, grant aims, ethics, methodology	Describe grant rationale, aims, and human subjects research. Explain key aspects of participant engagement, retention, and relationship. Explain ethical photography.	Group Discussion Written Reflection
Day 2: 3 h	Detail photovoice implementation plan	Describe photovoice implementation procedure. Demonstrate proficiency in describing the process and tasks to others.	Group Discussion Written Reflection Role play
Day 3: 3 h	Myth busting, use of camera, troubleshooting	Explain some misconceptions about aging and working with older adults. Practice using the camera, including troubleshooting.	Group discussion Role play

## Data Availability

The data are not publicly available due to data restriction policies.

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
