# Peer review of "“Pandemic Fatigue! It’s Been Going On since March 2020”: A Photovoice Study of the Experiences of BIPOC Older Adults and Frontline Healthcare Workers during the Pandemic"

_healthcare, 2022, doi:10.3390/healthcare10101967_

Round 1
Reviewer 1 Report
This qualitative study can be seen as a meaningful study for the health management of vulnerable people. However, some modifications are required in the research methodology as follows.
<Materials and Methods>
1. In qualitative research, research participants are very important because research participants are research tools. Describe the selection criteria and exclusion criteria of the study participants based on the rationale.
2. In this qualitative study, a total of 12 study participants were finally selected. Describe this on a theoretical basis. Describe whether the number of 12 study participants can be viewed as the number of samples that can secure representation for the population of this study.
3. Among the study participants, the age of Frontline healthcare workers is 20 to 30. If there is a reason for that, describe it.
4. Describe the research problem of this qualitative study.
Reviewer 2 Report
“Pandemic fatigue! It’s been going on since March 2020”: A Photovoice study of the experiences of BIPOC older adults and frontline healthcare workers during the pandemic. by Ekwonye et al.
To the Authors:
General comments:
The authors investigated the experiences of older adults and frontline healthcare workers during the COVID-19 pandemic through a qualitative study using photovoice methodology. It was considered that the study was well written, and the result included novelty. However, several points should be addressed to improve the manuscript.
Specific comments:
1. Did participants have medical histories of COVID-19? If there were participants who experienced COVID-19, what were the severities? This conceivably affects the interpretation of the results.
2. In lines 190-192, Institutional Review Board Statement was written; however, the approval number was missing. Please check that.
3. In Figures 6, 7, and 10, participants themselves were shown in photographs. Please mention who took these photographs. In procedure section (2.4), these situations seemed not to be anticipated. Also, did the authors obtain informed consent regarding the use of photographs showing their own faces?
Minor comments:
1. In the Abstract, the authors should spell out abbreviations, including “COVID-19” and “BIPOC” at the first appearance. Also, “COVID-19” and “COVID” are mixed; abbreviations should be unified into one.
2. In lines 168 and 171, “research assistants” was not abbreviated as “RA”. Please check that.
3. In line 332, “CAN” should be spelled out.
4. In line 388, “Covid” was mixed with capital and small characters. This may be a typo. Please check that.
Round 2
Reviewer 2 Report
“Pandemic fatigue! It’s been going on since March 2020”: A Photovoice study of the experiences of BIPOC older adults and frontline healthcare workers during the pandemic. by Ekwonye et al.
To the Authors:
General comments:
The authors incorporated changes to reflect the suggestions provided; however, there are still several points that should be addressed to improve the manuscript.
Minor comments:
1. The authors should spell out BIPOC in the Abstract, as I mentioned in the previous review. Also, "COVID-19" should be spelled out at the first appearance in the main text. Please follow Instructions for Authors: Acronyms/Abbreviations/Initialisms should be defined the first time they appear in each of three sections: the abstract; the main text; the first figure or table.
2. As I mentioned in the previous review, “research assistants” was not abbreviated as “RA” in line 196. Please check that.
3. Regarding Certified Nursing Assistant (CNA) in lines 361-362, I do not see the reason to show abbreviation. Please check that.
Author Response
General comments:
The authors incorporated changes to reflect the suggestions provided; however, there are still several points that should be addressed to improve the manuscript.
Minor comments:
1. The authors should spell out BIPOC in the Abstract, as I mentioned in the previous review. Also, "COVID-19" should be spelled out at the first appearance in the main text. Please follow Instructions for Authors: Acronyms/Abbreviations/Initialisms should be defined the first time they appear in each of three sections: the abstract; the main text; the first figure or table.
Response 1: The authors have addressed this concern.
2. As I mentioned in the previous review, “research assistants” was not abbreviated as “RA” in line 196. Please check that.
Response 2: The term research assistants have been abbreviated in relevant places.
3. Regarding Certified Nursing Assistant (CNA) in lines 361-362, I do not see the reason to show abbreviation. Please check that.
Response 3: The authors have removed the abbreviation (CNA).